# Diagnosis, Management and Prescription Practices of Adrenaline in Children with Food-Induced Anaphylaxis: Audit in a Specialized Pediatric Allergy Department

**DOI:** 10.3390/jpm12091477

**Published:** 2022-09-09

**Authors:** Konstantinos Vamvakaris, Alkmini Koumpoura, Maria Farmaki, John Lakoumentas, Maria Pasioti, Nikolaos Papadopoulos, Paraskevi Xepapadaki

**Affiliations:** 1Allergy Department, 2nd Pediatric Clinic, National and Kapodistrian University of Athens, Thivon & Levadeias St., Goudi, GR-11527 Athens, Greece; 2Division of Infection, Immunity and Respiratory Medicine, University of Manchester, Manchester M13 9PL, UK

**Keywords:** clinical audit, guidelines, food allergy, anaphylaxis, children, adrenaline

## Abstract

In the era of evidence-based medicine, physicians worldwide should abide by universally approved practices and healthcare units should seek quality control and operational improvement. This audit evaluates the degree of compliance with the European Academy of Allergy and Clinical Immunology guidelines for the diagnosis and treatment of anaphylaxis in a pediatric Allergy Department. Medical records of 248 children with food-induced allergic reactions who presented both on emergency and outpatient basis were reviewed. Data were also collected from the e-prescription database and anaphylaxis severity was graded according to Sampson’s criteria. An accuracy metric was used to calculate the consistency rate. Anaphylaxis was documented in 188/423 allergic reactions. The degree of agreement for the classification of the reactions as anaphylactic was 88.3%, while the respective rate for non-anaphylactic was 58.7%. In the anaphylactic cases, adrenaline was prescribed in 84.8%, while the respective rates for other drugs were: antihistamines: 27.6%; corticosteroids: 26.1%; inhaled β2-agonists: 11.8%. This study, through the example of pediatric food-induced anaphylaxis, underlines the significance of compliance to guidelines, organized documentation in healthcare units using specially formulated medical history forms and continuous medical stuff training. Thus, diagnosis and treatment practices can be improved for the benefit of patients.

## 1. Introduction

Clinical audits are a necessary process of quality control and, consequently, validation, certification of excellence and operational improvement in health institutions, that aim to provide patients with optimal diagnosis and treatment. According to the “Good Medical Practice” guidance of the General Medical Council, doctors are expected to take part in audits of their work as a means of promoting patient safety; thus, implementing changes to address any problems and/or carrying out further training if applicable [1].

The audit procedure includes the systematic review of clinical processes against explicit criteria, international recommendations and guidelines, in a methodical manner. The steps involved in a clinical audit are preparation and planning, measuring the level of performance, making improvements and maintaining improvements. More specifically, the first step includes the selection of a high priority topic that needs to be addressed, the identification of available resources and the definition of the gold standard. Following this step, collection and analysis of data is performed, evaluating compliance with the gold standard. Afterwards, suggested changes, where necessary, are discussed and applied by the relevant teams in the department, while the monitoring process should be repeated to ensure maximum improvement of the healthcare services offered [2,3,4,5].

Anaphylaxis is a severe, potentially fatal, systemic hypersensitivity reaction, most commonly IgE-mediated, with diverse clinical manifestations. The first-line therapeutic approach in anaphylactic cases includes ensuring airway patency, respiratory and circulatory support and immediate administration of intramuscular adrenaline. The second-line treatment includes oral or intravenous antihistamines, inhaled β2-agonists in case of bronchoconstriction and corticosteroids (intravenous or per os) [6,7]. Nevertheless, previous studies have shown that adrenaline is provided in a limited number of anaphylactic cases, ranging from 5–50%, according to the health setting. Moreover, patients frequently do not receive second-line therapy, and instructions for future management of an anaphylactic reaction are incomplete or even absent [8].

### 1.1. Objective

The purpose of this Audit was to evaluate:The degree of agreement with the international guidelines of cases characterized as anaphylactic, by reviewing recorded data in the medical files [8];The instructions and the prescribed medication provided.

#### 1.1.1. Primary Outcome

The primary outcome is the degree of agreement of the process followed for diagnosing a reaction as anaphylactic, based on the clinical signs and symptoms recorded in the patients’ medical charts, with the European Academy of Allergy and Clinical Immunology (EAACI) anaphylaxis definition.

#### 1.1.2. Secondary Outcome

The secondary outcome is the evaluation of coherence with the EAACI guidelines when it comes to the prescription of drugs for the treatment of a new anaphylactic reaction, based on severity grading according to Sampson’s criteria [6,9].

## 2. Materials and Methods

Medical records from children (up to the age of 12 years old) who presented to the Department either as a regular appointment or as an emergency, due to IgE-mediated reaction and/or anaphylaxis within two hours after consuming a food allergen were reviewed. The Allergy Department of the 2nd Pediatric Clinic in the University of Athens is the largest referral center for pediatric allergy and asthma in Greece, and consists of the outpatient clinic, which receives over 5000 allergic disease-associated patients yearly, certified as an excellence center in Pediatric Allergy (GA2LEN, UCARE, ADCARE). Moreover, anaphylactic reactions during oral food provocation performed in the Department for diagnostic reasons were included. Non-IgE reactions, such as proctocolitis and Food Protein-Induced Enterocolitis Syndrome (FPIES), and mixed reactions, such as atopic dermatitis and eosinophilic gastrointestinal disorders, were excluded [10]. Each reaction to a single food allergen provided a separate recording; thus, a patient with multiple food allergies would correspond to several recordings.

Anaphylaxis was defined, according to the most recent EAACI anaphylaxis guidelines, as a severe, potentially life-threatening hypersensitivity reaction including: (1) acute onset with skin and/or mucosal manifestations and at least one of the following: a. respiratory symptoms, e.g., dyspnea, wheezing, or b. drop in blood pressure with accompanying symptoms such as loss of consciousness and dizziness; (2) two or more of the following after exposure to a potential allergen: a. skin and/or mucous membrane symptoms, b. respiratory symptoms, c. symptoms from the cardiovascular system—drop in blood pressure or d. persistent gastrointestinal symptoms; and (3) blood-pressure drop after exposure to an already known allergen [8,11].

In addition, anaphylactic reactions were classified per severity using Sampson’s scale (Grade 1–5) to assess the documented management plan provided either in an emergency or outpatient setting. In our Department, following thorough evaluation of an anaphylactic reaction, we provide the patients with a written plan (and provide education) regarding any potential next allergic/anaphylactic incident that might happen at home (outside the hospital), including an epi-pen (auto-injector adrenaline) as first-line treatment and antihistamines and corticosteroids as second-line treatment. Adrenaline can be prescribed by the “emergency” doctors only for the first episode, while subsequent prescriptions must be provided according to national rules from specialized doctors (allergists) [9].

The patients’ files were initially evaluated by the first reviewer and reassessed by a second one (both medical students) in a course of 8 months. Any queries were discussed with an independent specialized doctor of the Department.

The data analyzed were: a. the type of the implicated food: milk, eggs, wheat, peanuts, nuts, fish, seafood, legumes, fruit, vegetables or other; b. age at reaction: children were grouped as preschoolers, 0–6 years old, or school aged, 7–12 years old; c. symptoms per system as reported during the anaphylactic reaction: skin, respiratory, gastrointestinal, cardiovascular, central nervous system (Table 1); d. a clear statement of an anaphylactic diagnosis (Tables 2 and 3); and e. medication prescribed for a potential subsequent anaphylactic reaction, including adrenaline, antihistamines, corticosteroids and β2-agonists (Table 4).

Symptoms and prescribed medication were categorized as “yes”, in case of a clear report in the file, “no”, in the event of absence of registration, and “incomplete data”, in case of absence of any report related to the specified parameter.

Regarding anaphylaxis diagnosis, a clear documentation had to be reported in the file.

### Statistical Analysis

Results are displayed in absolute number and proportion (%) of records. The accuracy metric was used to calculate the consistency rate, reflecting the degree of agreement with the gold standard, that is, the guidelines for diagnosing as well as treating anaphylaxis. We use the term “consistency rate” in order to quantify the degree of compliance with the EAACI guidelines, i.e., the compliance of our records compared to the “gold standard” defined by the clinical experts of the field. In order to perform this, we borrow the definition of the “accuracy metric” from machine learning. The “consistency rate”, measured with the “accuracy metric”, is a number in the [0, 1] interval (or the corresponding percent), generated by dividing the number of the correctly recorded instances (pointed out in bold font) to the total number of instances. For example, in Tables 2–4, it is the sum of the numbers in bold divided by the sum of all numbers and multiplied by 100.

## 3. Results

The retrospection of 248 medical files (of patients having presented during 2002–2020) provided 423 registrations, since reaction to a single allergen was documented as a separate registration. In total, 228 (91.9%) preschoolers and 20 (8%) school-aged children were included in the analysis. The following culprit foods were reported, in order of frequency: dairy products in 104 children (24.5%), nuts in 76 (17.9%), eggs in 66 (15.6%), fish in 58 (13.7%), legumes in 33 (7.8%), fruit in 19 (4.4%), wheat in 15 (3.5%), vegetables in 13 (3%), peanuts in 7 (1.6%), seafood in four (0.9%) and other food in 28 (6.6%).

The most commonly recorded symptom was urticaria/erythema in 329 (77.7%) of the reactions, whereas the respiratory system (shortness of breath, wheezing or hypoxemia) was involved in 166 (39.2%) cases. Angioedema was presented in 116 (27.4%), gastrointestinal symptoms (persistent abdominal pain and vomiting) in 169 (39.9%) and a drop in blood pressure (hypotension, dizziness or syncope) in 21 (4.9%) cases.

Incomplete symptoms recording was observed in 1.5% of the medical files. In 356 (84.2%) cases, there was no blood-pressure measurement at the time of the reaction (Table 1).

### 3.1. Documentation of an Anaphylactic Reaction

For 188/423 (44.4%) reactions, there was a clear report of anaphylaxis in the medical file, while in the 235 remaining ones (55.5%) there was no report.

In four out of 188 (2.1%) reactions characterized as anaphylactic, no symptom recording was evident in the files, while in 20 (10.6%) there were symptoms concerning only one system; however, two of them involved the cardiovascular system, which was sufficient for classifying the reaction as anaphylactic. In the remaining 164 cases, there were reports of symptoms from two systems: 84 (44.7%); three systems: 57 (30.3%); or more: 23 (12.2%). The degree of agreement regarding the characterization of the reactions as anaphylactic was 166/188 (88.3%).

From the 235 reactions for which there was no definitive characterization as non-anaphylactic, eight (3.4%) had no symptoms reported; 130 (55.3%) had recorded symptoms from one system; 74 (31.4%) from two; 22 (9.3%) from three; and 1/235 (0.4%) from four systems. Therefore, the degree of agreement with regard to characterization as non-anaphylactic was 138/235 (58.7%). In total, 97 reactions fulfilled the criteria of anaphylaxis, however, no clear report was provided (Table 2).

### 3.2. Severity Grading of Anaphylactic Reactions

Severity grading according to Sampson’s criteria was applied for 175 out of 188 anaphylactic reactions, since data for 13 reactions were either absent or insufficient (Appendix A). The severity of the reactions was classified as follows: Grade 1: 0; Grade 2: 36 (19.5%); Grade 3: 50 (27.2%); Grade 4: 84 (46.6%); and Grade 5: five (2.7%). The degree of agreement between the medical file reports and the audit’s grading was 175/188 (93%). From the 235 reactions characterized as non-anaphylactic in the medical files, 206 did not fulfil the anaphylaxis criteria in terms of severity, whereas for 29 there was no explicit anaphylaxis diagnosis, despite their classification according to Sampson’s criteria by the audit as following: Grade 1: 1 (0.4%); Grade 2: 18 (7.6%); Grade 3: 5 (2.1%); and Grade 4: 5 (2.1%). Thus, the degree of agreement was 206/235 (87.6%) (Table 3). Using severity grading as a secondary criterion, a total of 204 (175 and 29) reactions could have been recorded as anaphylactic.

### 3.3. Anaphylaxis Management Plan Provided for a Future Episode

According to the audit’s criteria for managing future anaphylactic reactions based on severity grading, the rates for prescribing first- and second-line treatment for home use in the 204 anaphylactic reactions were as follows. (a) Adrenaline was prescribed in 173 (84.8%) cases. (b) Antihistamines were prescribed in 56 (27.6%) anaphylactic reactions and in 35/215 (16.3%) non-anaphylactic reactions. (c) Corticosteroids were prescribed in 53 (26.1%) of the anaphylactic reactions and in 33/215 (15.3%) non-anaphylactic reactions. (d) β2-agonists were prescribed in 24 (11.8%) anaphylactic reactions and in 15/215 (7%) non-anaphylactic reactions (Table 4). In 19 children we found a prescription of adrenaline in the electronic records of the national prescription system, although no such data were recorded in their medical files. In reactions not classified as anaphylactic (215) by the audit’s authors, adrenaline was prescribed in 114/215 (53%).

## 4. Discussion

In the present audit, the overall consistency rate regarding the compliance with the EAACI definition of anaphylaxis was 71.39%, while the consistency rate concerning the characterization of all the reactions based on severity (according to Sampson’s criteria) was 90.93%. In 29 reactions classified as Grade 1–4, a clear diagnosis of anaphylaxis was not reported in the medical file. Underestimation of cases as non-anaphylactic could be attributed to either mild manifestations from specific systems (e.g., gastrointestinal) or to the remission of relevant symptoms at the time of clinical examination during the incident.

Regarding the medication plan provided for future management of new episodes, prescription of adrenaline was recorded in the majority (84.8%) of the patients’ files. Additionally, 19 children were prescribed adrenaline, as was indicated by search in the e-prescription national database, which was not recorded in their medical files though. This means that despite that a proper treatment plan was provided, a lack in the documentation strategy of the Unit is noticed. Furthermore, adrenaline injectors were prescribed for 114 cases which did not meet the requirements for being characterized as anaphylactic reactions. This may be attributed to the severity of the incident, even in the absence of symptoms from multiple systems, to strong clinical suspicion (based on relevant literature) of the occurrence of future anaphylaxis or to parental anxiety. Additionally, it is known that Immunoglobin E-mediated allergies may develop tolerance over time and patients may exhibit milder clinical presentation or, on the other hand, patients may present with more severe symptoms at lower doses of an allergen in the presence of facilitating factors [12,13]. In addition, it is well established that patients with uncontrolled asthma present with more severe forms of food-dependent anaphylactic reactions [14]. Although we did not assess the prevalence of other allergy-associated diseases in our cohort, a proportion of patients did not present a personal history compatible with asthma.

Skin symptoms were the most prevalent reported symptoms during anaphylactic reactions, while cardiovascular involvement was recorded in a relatively small percentage of the reactions (5%). In the study of Worm et al., which included 2012 pediatric and adult patients, skin symptoms also prevailed with a percentage of 83.9%, whereas cardiovascular manifestations were significantly more common (71.7%) [15]. Moreover, a lower rate of blood-pressure measurement in cases of pediatric anaphylaxis was noted, compared to the rate of 75% in the Mclaughlin study [16]. The limited percentage of blood-pressure measurement during the event is probably due to the inevitably delayed physical examination by the doctors, since most reactions occur in an ambulatory context [11].

In a retrospective review of 20 anaphylaxis cases in children at a general regional hospital, while sufficient pre-hospital adrenaline administration during the reaction (85%) was reported, the prescription rates for an auto-injector at discharge were only 70% [16]. In the Murad and Katelaris study, in 58 cases (children and adults) who visited the emergency department of a tertiary referral hospital, 54% of the patients received adrenaline during the first anaphylactic reaction, while in 75% of them, adrenaline was provided for the treatment of subsequent reactions [17]. Moreover, in an audit including data from 146 adults with an anaphylactic reaction, only 43% of the patients received adrenaline, either as monotherapy or in combination with antihistamine or hydrocortisone, after emergency anaphylaxis diagnosis [18]. According to data obtained from emergency departments, the prescription and the administration of adrenaline are often delayed [17].

Emergency administration of corticosteroids and antihistamines during anaphylactic reactions is reported at a range of 48–72% in previous studies [18,19]. In the present audit, corticosteroids and antihistamines were prescribed as a management plan in 26.1% and 27.6% of the anaphylactic cases respectively.

The study shows high compliance with guidelines in respect to the anaphylaxis diagnosis, although in a small number of reactions characterized as anaphylactic (4/423) there was evidence only for skin symptoms, which may be due to under- or incomplete recording of other symptoms. In addition, during the time lapse between the onset of each episode and the time of clinical examination, symptoms may subside, or the parents may not be able to accurately recall them. Deviations from accurate anaphylaxis diagnosis may be attributed to the absence of a simple and easily memorable definition of anaphylaxis and the lack of a specifically designated form [20].

## 5. Limitations/Drawbacks of the Study

The data presented may be due to the lack of efficient documentation and not necessarily to non-compliance with the established guidelines. This is explained by the design of this study, which is based on a retrospective review of the medical files by independent observers.

## 6. Conclusions

This is the first reported audit aiming to evaluate health services provision in the pediatric population with reported IgE-mediated food-induced reactions in our country. We conclude that diagnosis of anaphylaxis according to the international guidelines was satisfactory (88.3%), while prescription of first-line treatment for future anaphylactic reactions was noted in 84.8% of the cases.

It is suggested that a structured, specifically designated form for cases with an anaphylactic reaction should be used, where the symptoms will be clearly stated per system, and the diagnosis (or not) of anaphylaxis and the management plan will be filled in in the respective fields (Appendix A).

Thus, by improving documentation and establishing continuous education of healthcare providers, better compliance to universally approved guidelines can be achieved, ensuring the efficiency of any medical department. Data deriving from this project in a specialized pediatric center highlight the necessity for quality assessment using defined criteria. As a result, the audit process constitutes a valuable tool in the hands of physicians aiming to elucidate errors and omissions in healthcare provision and a means to assist in better communication amongst a department’s personnel.

## Figures and Tables

**Table 1 jpm-12-01477-t001:** Recording of symptoms per system in accordance with the reaction’s severity.

		Anaphylactic Reaction Severity							
		Incomplete Data	No Anaphylaxis	Grade 1	Grade 2	Grade 3	Grade 4	Grade 5	Total
Sudden onset of skin symptoms	Yes	0 (0%)	165 (76.74%)	1 (100%)	47 (87.04%)	42 (76.36%)	69 (77.53%)	5 (100%)	329 (77.78%)
	No	1 (25%)	48 (22.33%)	0 (0%)	7 (12.96%)	13 (23.64%)	19 (21.35%)	0 (0%)	88 (20.8%)
	Incomplete Data	3 (75%)	2 (0.93%)	0 (0%)	0 (0%)	0 (0%)	1 (1.12%)	0 (0%)	6 (1.42%)
Sudden onset of angioedema	Yes	0 (0%)	50 (23.26%)	0 (0%)	17 (31.48%)	20 (36.36%)	28 (31.46%)	1 (20%)	116 (27.42%)
	No	1 (25%)	161 (74.88%)	1 (100%)	37 (68.52%)	35 (63.64%)	60 (67.42%)	4 (80%)	299 (70.69%)
	Incomplete Data	3 (75%)	4 (1.86%)	0 (0%)	0 (0%)	0 (0%)	1 (1.12%)	0 (0%)	8 (1.89%)
Respiratory symptoms	Yes	0 (0%)	19 (8.84%)	0 (0%)	23 (42.59%)	37 (67.27%)	83 (93.26%)	4 (80%)	166 (39.24%)
	No	1 (25%)	195 (90.7%)	1 (100%)	31 (57.41%)	17 (30.91%)	6 (6.74%)	1 (20%)	252 (59.57%)
	Incomplete Data	3 (75%)	1 (0.47%)	0 (0%)	0 (0%)	1 (1.82%)	0 (0%)	0 (0%)	5 (1.18%)
Blood pressure drop	Yes	0 (0%)	3 (1.4%)	0 (0%)	5 (9.26%)	1 (1.82%)	8 (8.99%)	4 (80%)	21 (4.96%)
	No	0 (0%)	16 (7.44%)	0 (0%)	6 (11.11%)	9 (16.36%)	15 (16.85%)	0 (0%)	46 (10.87%)
	Incomplete Data	4 (100%)	196 (91.16%)	1 (100%)	43 (79.63%)	45 (81.82%)	66 (74.16%)	1 (20%)	356 (84.16%)
Persistent gastrointestinal symptoms	Yes	0 (0%)	50 (23.26%)	1 (100%)	37 (68.52%)	41 (74.55%)	38 (42.7%)	2 (40%)	169 (39.95%)
	No	1 (25%)	161 (74.88%)	0 (0%)	17 (31.48%)	14 (25.45%)	50 (56.18%)	3 (60%)	246 (58.16%)
	Incomplete Data	3 (75%)	4 (1.86%)	0 (0%)	0 (0%)	0 (0%)	1 (1.12%)	0 (0%)	8 (1.89%)
Total symptoms	Yes	0 (0%)	287 (26.7%)	2 (40%)	129 (47.78%)	141 (51.27%)	226 (50.79%)	16 (64%)	801 (37.87%)
	No	4 (20%)	581 (54.05%)	2 (40%)	98 (36.3%)	88 (32%)	150 (33.71%)	8 (32%)	931 (44.02%)
	Incomplete Data	16 (80%)	207 (19.25%)	1 (20%)	43 (15.92%)	46 (16.73%)	69 (15.5%)	1 (4%)	383 (18.11%)

**Table 2 jpm-12-01477-t002:** Recording of “Anaphylaxis” diagnosis according to the number of systems involved.

		A Clear Statement of an Anaphylactic Reaction from Medical Files	
		No	Yes
Number of systems from which symptoms were manifested	0	**8**	4
	1	**130**	20
	2	74	**84**
	3	22	**57**
	4	1	**21**
	5	0	**2**
Total		235	188
	Consistency Rate (%):		71.39

**Table 3 jpm-12-01477-t003:** Recording of “Anaphylaxis” diagnosis according to the reaction’s severity (based on Sampson’s criteria).

		A Clear Statement of an Anaphylactic Reaction from Medical Files	
		No	Yes
Reaction’s severity	No anaphylaxis	**206**	9
	Grade 1	1	**0**
	Grade 2	18	**36**
	Grade 3	5	**50**
	Grade 4	5	**84**
	Grade 5	0	**5**
	Consistency Rate (%):		90.93

**Table 4 jpm-12-01477-t004:** Rates of medication provided for future anaphylactic reactions, based on the reaction’s severity.

		Adrenaline		Antihistamines		Cortisone		β_2_-Agonist	
		No	Yes	No	Yes	No	Yes	No	Yes
Reaction’s severity	No anaphylaxis	**101**	114	**180**	35	**182**	33	**200**	15
	Grade 1	1	**0**	1	**0**	1	**0**	1	**0**
	Grade 2	13	**41**	43	**11**	45	**9**	51	**3**
	Grade 3	9	**46**	32	**23**	33	**22**	42	**13**
	Grade 4	8	**81**	68	**20**	68	**20**	80	**8**
	Grade 5	0	**5**	3	**2**	3	**2**	5	**0**
Consistency Rate (%):		65.39		56.46		56.22		53.83	

## Data Availability

The raw data presented in this article and supporting the conclusions are available upon reasonable request from the corresponding author.

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
