# Peer review of "Diagnosis, Management and Prescription Practices of Adrenaline in Children with Food-Induced Anaphylaxis: Audit in a Specialized Pediatric Allergy Department"

_jpm, 2022, doi:10.3390/jpm12091477_

Round 1

Reviewer 1 Report

Overall an interesting article, but some issues need clarification. I have listed my suggestions below. 

 Introduction

Introduction needs improvement. Audit should be explained, not a very common research method for pediatric allergists.

Material methods

I suggest some additions to this section.

Brief information about the department where the study was conducted (such as bed capacity, outpatient density, etc.)

Time period of the study.

Do emergency or allergy outpatient clinics perform the prescribing of adrenaline autoinjector in your unit?.

Line 101-105

Please explain what is "accuracy metric" and "consistency rate" specify how it is calculated,

Specify if there are patients who have had anaphylaxis during oral provocation.

Is there a patient with multiple food allergies?

Results

Result part needs to be rearranged

Discussion

Indicate that Immunoglobulin E-mediated food allergy may develop tolerance over time and patients may tolerate the food, or that more severe reactions may occur at lower doses in the presence of facilitating factors. Also, emphasize that some of the patients with food allergies may not have asthma and that adrenaline injectors can be prescribed even if the reaction is not anaphylactic.

 The literature is old and few in number, current literature should be searched.

Author Response

Comments from reviewer 1

Introduction

Introduction needs improvement. Audit should be explained, not a very common research method for pediatric allergists.

Response: Information about the steps of an Audit process, as well as explanation of why audits are a necessary part of good clinical practice is now provided in the section “1. Introduction”. (lines: 35-38, 41-46)

Materials and methods

I suggest some additions to this section.

Brief information about the department where the study was conducted (such as bed capacity, outpatient density, etc.)

Response:  We have added information about our Allergy Department’s capacity and personnel in the first paragraph of the section “2. Materials and Methods”. (lines: 77-80)

Time period of the study.

Response: The time period of our study was 8 months and is now clearly stated in the fourth paragraph of the section “2. Materials and Methods”. (line: 106)

Do emergency or allergy outpatient clinics perform the prescribing of adrenaline autoinjector in your unit?

Response: In our Audit, we included both patients who were examined in the emergency department and on a regular/outpatient basis. In the Department adrenaline auto-injector is prescribed in any of these two settings. Adrenaline can be prescribed by the “emergency” doctors only for the first episode, while subsequent prescriptions must be provided according to national rules from specialized doctors (allergists). A relevant comment is provided in the third paragraph of the section “2. Materials and Methods”. (lines: 97-104)

Line 101-105

Please explain what is "accuracy metric" and "consistency rate" specify how it is calculated,

Response: Thank you for this thoughtful comment. Indeed, these terms were not adequately explained. We use the term "consistency rate" in order to quantify the degree of compliance with the EAACI guidelines; i.e., the compliance of our records compared to the "gold standard" defined by the clinical experts of the field. In order to perform this, we borrow the definition of the "accuracy metric" from machine learning. The "consistency rate", measured with the "accuracy metric", is a number in the [0,1] interval (or the corresponding percent), generated by dividing the number of the correctly recorded instances to the total number of instances. For example, in Tables 2, 3 and 4, it is the sum of the numbers in bold divided by the sum of all numbers, and multiplied with 100. We have elaborated on the definitions of the two terms and given analytical information about their use and calculation in the section “2.1 Statistical Analysis” (lines: 124-131)

Specify if there are patients who have had anaphylaxis during oral provocation.

Response: Patients who experienced allergic or anaphylactic reactions during oral food provocation in our Department were also enrolled in the study. This is stated in the first paragraph of the section “2. Materials and Methods”. (lines: 80-82)

Is there a patient with multiple food allergies?

Response: Several patients presented allergic reactions to more than one food allergens. Therefore, more than one recordings could correspond to the same patient and as a result our study included 248 patients, but our recordings reached the number 423. This is specified in the first paragraph of “2. Materials and Methods” (lines: 84-86) and in the first paragraph of the section “3. Results” (lines: 134-135)

Results

Result part needs to be rearranged

Response: We have somehow rearranged/rephrased results, providing more detailed subtitles in order to increase clarity.

Discussion

Indicate that Immunoglobulin E-mediated food allergy may develop tolerance over time and patients may tolerate the food, or that more severe reactions may occur at lower doses in the presence of facilitating factors.

Response: Thank you for this insightful addition. We have provided an explanation concerning this issue in the second paragraph of the section “4. Discussion” (lines: 212-215)

Also, emphasize that some of the patients with food allergies may not have asthma and that adrenaline injectors can be prescribed even if the reaction is not anaphylactic.

Response: We agree that your comment is correct. However, we did not assess the prevalence of other allergy-associated diseases. In respect to the fact that adrenaline might be prescribed even in the absence of a clear history of anaphylactic reaction, this was also the case in our study and we provide a relevant statement in the second paragraph of the section “4. Discussion” (lines: 216-219).

The literature is old and few in number, current literature should be searched.

Response: We have updated the references according to your suggestion. References 1-4, 7, 11-14, 20 are now added in the list.

Reviewer 2 Report

This is an interesting study of the compliance/adherence to the guidelines for anaphylaxis regarding diagnosis and treatment.  I have a few comments on clarity of presentation:

1.  The title should state that this is a study of pediatric patients in a clinic/emergency department/allergy clinic (need to clarify) and that you evaluated documentation of diagnostic criteria, treatment of anaphylaxis and home epinephrine/adrenaline.

2.  The abstract should more clearly state where the patients were evaluated.  Was this a clinic or did the allergists go to the emergency room, or both?  The paper references the allergy department, but I think the specifics are necessary because allergists will be held to a higher standard and comparison of other studies and prescribing habits of other specialties will be of interest.

3. It is not clear to me what you mean by 248 files were evaluated but 423 registrations.  Do you mean some patients had multiple episodes of anaphylaxis?

4.  Just for clarity I believe you are stating that 188 cases of 423 were diagnosed as anaphylaxis.  175 of the 188 were verified as anaphylaxis by review and some additional cases that were not identified as anaphylaxis by the providers  brought the number of verified cases to 204.  423 cases were listed as acute allergic reactions to food in children, and some of these cases occurred during food challenge.  Is this correct?  The 93% agreement for diagnosis is good, and it would be interesting to compare to ER/generalist data.

5.  Could you please elaborate on the 19 patients with anaphylaxis who did not receive adrenaline treatment by documentation but escribe data differed?  Do you mean the provider did not say that gave epinephrine in their note or do you mean they already had an epi-pen at home?  Confused.

6.  I find some confusion on presentation of data on adrenaline prescribed for acute episode and adrenaline prescribed for home use.

7.  Please elaborate on how this data could be utilized as teaching tool for physicians

Overall I find this data very interesting.  Thank you

Author Response

Comments from reviewer 2

  1. The title should state that this is a study of pediatric patients in a clinic/emergency department/allergy clinic (need to clarify) and that you evaluated documentation of diagnostic criteria, treatment of anaphylaxis and home epinephrine/adrenaline.

Response: The title has been modified according to your suggestions. Our cohort includes children who were evaluated both in the emergency department and as outpatients, so we have summed it up in the phrase “Allergy Department”.   

  1. The abstract should more clearly state where the patients were evaluated. Was this a clinic or did the allergists go to the emergency room, or both?  The paper references the allergy department, but I think the specifics are necessary because allergists will be held to a higher standard and comparison of other studies and prescribing habits of other specialties will be of interest.

Response: Our study design entailed patients who presented both on emergencies where the allergists evaluated the children and at the Department on an outpatient basis. The Unit’s doctors were called to the Emergency Department in case of an allergy/anaphylaxis case, or/and patients were subsequently evaluated at a prescheduled appointment. The documentation of the medical records and follow-up is the same for all the patients according to international guidelines. The Abstract has been updated accordingly.

  1. It is not clear to me what you mean by 248 files were evaluated but 423 registrations.  Do you mean some patients had multiple episodes of anaphylaxis?

Response: A substantial number of patients presented allergic reactions to more than one food allergens. Therefore, more than one recording correspond to the same patient and, as a result, our study included 248 patients, however they presented distinct reactions (cases) up to 423. This is specified in the first paragraph of “2. Materials and Methods”. (lines: 84-86) and in the first paragraph of the section “3. Results” (lines: 134-135)

  1. Just for clarity I believe you are stating that 188 cases of 423 were diagnosed as anaphylaxis. 175 of the 188 were verified as anaphylaxis by review and some additional cases that were not identified as anaphylaxis by the providers brought the number of verified cases to 204. 423 cases were listed as acute allergic reactions to food in children, and some of these cases occurred during food challenge. Is this correct? The 93% agreement for diagnosis is good, and it would be interesting to compare to ER/generalist data.

Response: We confirm that your understanding of the analysis is correct and in the third – fifth paragraph of section “4. Discussion” we have compared our findings to relevant previous studies. (lines: 222-243)

  1. Could you please elaborate on the 19 patients with anaphylaxis who did not receive adrenaline treatment by documentation but escribe data differed? Do you mean the provider did not say that gave epinephrine in their note or do you mean they already had an epi-pen at home? Confused.

Response: Thank you for your constructive remark. In the case of these 19 patients, there is a lack only in the documentation of adrenaline prescription in their medical files, since the search in the national electronic prescription database revealed that adrenaline had, indeed, been prescribed. An explanation for this is provided in the second paragraph of the section “4. Discussion”. (lines: 206-208)

  1. I find some confusion on presentation of data on adrenaline prescribed for acute episode and adrenaline prescribed for home use.

Response: In our Department, following thorough evaluation of an anaphylactic reaction, we provide the patients with a written plan (and provide education) regarding any potential next allergic/anaphylactic incident that might happen at home (outside the hospital), including an epi-pen (auto-injector adrenaline) as first-line treatment and antihistamines and corticosteroids as second-line treatment. In the hospital, the adrenaline that is used for acute episodes does not need to be prescribed. We have added information concerning the prescription of first- and second-line treatment for future episodes in the third paragraph of the section “Materials and Methods” (lines: 98-104) and also the phrase “for home use” in the section “3.3. Anaphylaxis management plan provided for a future episode” (line: 183)

  1. Please elaborate on how this data could be utilized as teaching tool for physicians

Response: We have provided further information in the section “6. Conclusions” on how the present study can be helpful as a means to underline the benefits of applying auditing in health institutions, so as to improve patient care. (lines: 271-273) Moreover, a specifically designated form for cases with an anaphylactic reaction was constructed, where the symptoms can be clearly stated per system, and the diagnosis (or not) of anaphylaxis and the management plan can be filled in in the respective fields. This Special History Form is now provided as Supplementary Material 2.